# Assessing Judging Bias in Large Reasoning Models: An Empirical Study

**Qian Wang, Zhanzhi Lou, Zhenheng Tang, Nuo Chen, Xuandong Zhao**

**Wenxuan Zhang, Dawn Song, Bingsheng He**

## Abstract

Large Reasoning Models (LRMs) like DeepSeek-R1 and OpenAI-o1 have demonstrated remarkable reasoning capabilities, raising important questions about their biases in LLM-as-a-judge settings. We present a comprehensive benchmark comparing judging biases between LLMs and LRMs across both subjective preference-alignment datasets and objective fact-based datasets. Through investigation of bandwagon, authority, position, and distraction biases, we uncover four key findings: (1) despite their advanced reasoning capabilities, LRMs remain susceptible to the above biases; (2) LRMs demonstrate better robustness than LLMs specifically on fact-related datasets; (3) LRMs exhibit notable position bias, preferring options in later positions; and (4) we identify a novel "superficial reflection bias" where phrases mimicking reasoning (e.g., "wait, let me think...") significantly influence model judgments. To address these biases, we design and evaluate three mitigation strategies: specialized system prompts that reduce judging biases by up to 19% in preference alignment datasets and 14% in fact-related datasets, in-context learning that provides up to 27% improvement on preference tasks but shows inconsistent results on factual tasks, and a self-reflection mechanism that reduces biases by up to 10% in preference datasets and 16% in fact-related datasets, with self-reflection proving particularly effective for LRMs. Our work provides crucial insights for developing more reliable LLM-as-a-Judge frameworks, especially as LRMs become increasingly deployed as automated judges. Our code is available at https://github.com/Persdre/LRM-bias-evaluation.

## 1 Introduction

As Large Language Models (LLMs) have demonstrated remarkable capabilities across many domains (Brown et al., 2020; Wei et al., 2022), researchers increasingly deploy them as automated evaluators—a paradigm known as Model-as-a-Judge (Gu & Others, 2024; Li & Others, 2024). Recently, LRMs such as DeepSeek-R1 (Guo et al., 2025) and OpenAI-o1 (OpenAI, 2025) have emerged, demonstrating superior performance in complex problem-solving tasks including mathematics and programming (Xu et al., 2025). These models incorporate structured reasoning mechanisms like chain-of-thought (Wei et al., 2023) and self-reflection (Madaan et al., 2023), offering enhanced accuracy and interpretability compared to LLMs. This advancement raises important questions about how reasoning capabilities might affect judging performance when these models serve as automated evaluators.

Traditional LLMs have been observed with various biases when used as automatic model judges (Ye et al., 2024). For instance, when serving as judges, LLMs exhibit position bias (Zheng et al., 2024), preferring answers based on their ordered position rather than content quality. Similarly, LLMs' judgments shown susceptibility to bandwagon effects during evaluation (Koo et al., 2023). While these judging biases have been studied in LLMs, to our knowledge, no work has examined how reasoning-enhanced LRMs might be affected by these same biases in evaluation or introduce new judging bias. Furthermore, recent studies suggest that LRMs are less robust than LLMs in certain safety aspects, as their longer

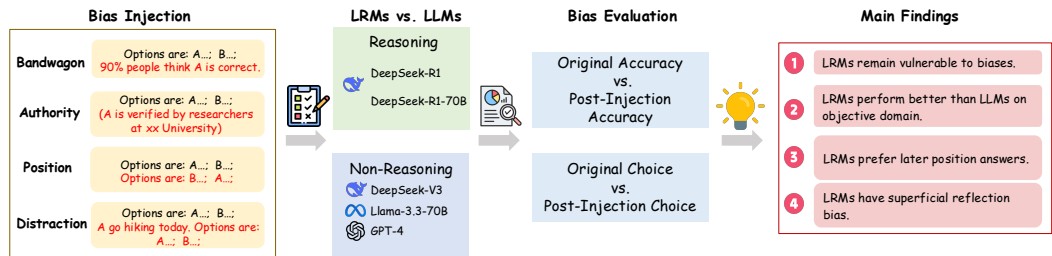

Figure 1: We develop a comprehensive framework to systematically evaluate judging biases across LLMs and LRMs, with three primary objectives: (1) assessing bias susceptibility in LRMs during evaluation tasks, (2) comparing judging bias patterns between LLMs and LRMs, (3) analyzing the formation of evaluation biases in LRMs' reasoning processes, and (4) identifying new judging biases in LRMs.

chain-of-thought processes create more vulnerability points for attacks (Zhou et al., 2025; Huang et al., 2025). These considerations motivate us to systematically investigate the following questions:

*How do LRMs perform when evaluating content as automated judges? What are the similarities and differences between LRMs and LLMs in judging reliability? How can we leverage enhanced reasoning mechanisms to mitigate cognitive biases when LRMs serve as automated evaluators?*

To answer these questions, we design a comprehensive benchmark to investigate judging bias patterns across LLMs and LRMs. As shown in Figure 1, our evaluation examines four critical cognitive biases in automated evaluation settings (Koo et al., 2023; Ye et al., 2024): bandwagon bias, authority bias, position bias, and bias under distraction. We evaluate models on both human preference alignment datasets (DPO datasets) (Leo, 2024; Intel, 2023; Durbin, 2024; 2023) and objective fact-related questions (Wang et al., 2024), comparing models within the same architectural families to isolate reasoning effects. We also analyze LRMs' intermediate reasoning steps (content between <think> and </think> tags) to understand bias formation mechanisms during evaluation.

We have four main findings from our experiments: **(1)** Despite their advanced reasoning capabilities, LRMs exhibit significant vulnerability to the aforementioned judging biases; **(2)** LRMs demonstrate greater robustness than LLMs when evaluating fact-related content; **(3)** When serving as judges, LRMs show a consistent preference for options appearing in later positions; and from **(3)** we identify **(4)** LRMs display a novel **"superficial reflection bias"** where simply inserting phrases like *"wait, let me think about it"* between options significantly increases preference for the later answer. These findings reveal that despite advanced reasoning capabilities, LRMs exhibit unique vulnerability patterns in judging, stemming from their training to prioritize reasoning-like text patterns.

Based on our benchmark and understanding of these judging bias mechanisms, we propose three complementary strategies to mitigate judging biases: (1) a specialized system prompt that explicitly targets previously identified evaluation vulnerabilities; (2) in-context learning (ICL) with examples demonstrating unbiased judging; and (3) a self-reflection mechanism that encourages models to critically evaluate their reasoning processes; Our experiments reveal that each strategy has distinct strengths: system prompts reduce judging biases by up to 19% in human preference alignment datasets and 14% in fact-related datasets; self-reflection reduces biases by up to 10% in preference alignment datasets and 16% in fact-related datasets; while ICL demonstrates the strongest performance on preference tasks with up to 27% improvement but shows inconsistent results on factual tasks. We find that self-reflection is particularly effective for LRMs, leveraging their stronger reasoning capabilities, while ICL provides greater benefits for LLMs on preference-based tasks. These complementary approaches represent promising directions for reducing judging biases across different model architectures and evaluation contexts.

We make the following contributions:

- We develop a comprehensive benchmark evaluating judging biases across LLMs and LRMs, revealing that LRMs remain susceptible to evaluation biases despite their reasoning capabilities, while showing improved robustness on fact-related content.

- We identify a novel "superficial reflection bias" in LRMs' evaluation processes, where phrases mimicking reasoning (e.g., "wait, let me think...") significantly influence judging outcomes, demonstrating how reasoning mechanisms can introduce new vulnerabilities in automated evaluation.

- We design and validate three simple and intuitive bias mitigation strategies: (1) specialized system prompts that reduce judging biases by up to 19% in preference alignment datasets and 14% in fact-related datasets, (2) in-context learning that provides up to 27% improvement on preference tasks but shows inconsistent results on factual tasks, and (3) a self-reflection mechanism that reduces biases by up to 10% in preference datasets and 16% in fact-related datasets, with self-reflection proving particularly effective for LRMs due to their stronger reasoning capabilities.

## 2 Judging Bias Evaluation Design

### 2.1 Judging Bias Evaluation Framework

We formalize the process of evaluating judgments produced by a judge model $M$, which can be a standard LLM or a LRM. Given a task instruction $I$ and an input query $Q$, the model $M$ evaluates a set of candidate items $\mathcal{R}$. The model's primary output is a final judgment $J = M(I, Q, \mathcal{R})$. While LRMs might generate intermediate reasoning $S$ and reflection $\Phi$, our quantitative analysis focuses on the final judgment $J$ and its derived score. We consider two primary evaluation formats:

**Pair-wise Comparison.** The set of candidates is $\mathcal{R} = \{R_A, R_B\}$, representing two distinct responses. The judgment $J$ indicates a preference relation between $R_A$ and $R_B$. We map this judgment to a binary score $y$:

$$y = \mathbf{1}(R_A \succ_J R_B) \in \{0, 1\} \tag{1}$$

where $R_A \succ_J R_B$ signifies that judgment $J$ prefers $R_A$ over $R_B$, and $\mathbf{1}(\cdot)$ is the indicator function. By convention, $y = 0$ implies $R_B \succ_J R_A$.

**Multiple-Choice Selection.** The set of candidates is $\mathcal{R} = \{O_1, \ldots, O_k\}$, representing $k$ distinct options. The judgment $J \in \mathcal{R}$ corresponds to the option selected by the model. Let $O^* \in \mathcal{R}$ denote the ground-truth correct option. We define the accuracy score $y$:

$$y = \mathbf{1}(J = O^*) \in \{0, 1\} \tag{2}$$

These definitions provide a unified quantitative score $y \in \{0, 1\}$ based on the model's judgment $J$ across different task formats.

### 2.2 Judging Bias Benchmark Design

**Comparing LLMs and LRMs.** To analyze whether bias susceptibility stems from model families or reasoning capabilities, we carefully select models that allow for controlled comparisons. We evaluate two LRMs: **DeepSeek-R1 (DS-R1)** (Guo et al., 2025), the strongest model in the R1 series; and **DeepSeek-R1-70b (R1-70b)**, a reasoning model distilled from Llama 3.3-70b (Guo et al., 2025). For comparison, we include three LLMs without explicit reasoning capabilities: **GPT-4o** (OpenAI, 2024), **Llama 3.3-70b (Llama3.3)** (Dubey et al., 2024), and **DeepSeek-V3 (DS-V3)** (Liu et al., 2024). This selection enables direct comparison between reasoning and non-reasoning variants from the same model families (DeepSeek-R1 vs. DeepSeek-V3, and Llama-distilled-R1 vs. Llama 3.3), allowing us to isolate the impact of reasoning capabilities on bias susceptibility.

**Comparing Human Preference Alignment v.s. Factual Datasets.** To investigate how LRMs behave differently when evaluating factual versus subjective content, we employ both subjective and objective benchmarking datasets: (1) Subjective DPO datasets (which contain

human-labeled preference pairs where one response is preferred over another): **Emerton-DPO** (Leo, 2024), **Orca-DPO** (Intel, 2023), **Py-DPO** (Durbin, 2024), and **Truthy-DPO** (Durbin, 2023); and (2) Objective fact-related datasets adapted from MMLU-Pro (Wang et al., 2024): **Math**, **Chemistry**, **History**, and **Psychology**, which contain multiple-choice questions (each question has 10 options) with factually correct answers. This dual-dataset approach allows us to examine whether reasoning mechanisms provide different levels of bias protection depending on the task type. Details are in Appendix A.1.

**Hyperparameters.** We set the temperature parameter to 0.7 for all models, consistent with the experimental settings established in prior work (Ye et al., 2024; Tan et al., 2024).

**Evaluation Metrics.** Building on our framework in Section 2.1, we evaluate models using two metrics: **Accuracy** and **Robustness Rate (RR)**. For each evaluation scenario, the model produces a judgment $y$ under normal conditions and a judgment $\hat{y}$ after bias injection. The ground truth is denoted as $y^*$. The metrics are defined as:

$$\textbf{Accuracy} = \frac{1}{|D|} \sum_i \mathbb{I}(y^i = y^{*i}), \quad \textbf{RR} = \frac{1}{|D|} \sum_i \mathbb{I}(y^i = \hat{y}^i).$$

where $|D|$ represents the size of the dataset. **Accuracy** measures how often the model's judgment $y$ correctly aligns with the ground truth $y^*$. **RR** quantifies consistency by measuring how often the model's judgment remains unchanged after bias injection. Note that for all experiments, we repeat three times and report the average results.

# 3 Judging Bias Benchmarking

## 3.1 Bandwagon Bias

| Model | Emerton-DPO | | | Orca-DPO | | | Py-DPO | | | Truthy-DPO | | |
|---|---|---|---|---|---|---|---|---|---|---|---|---|
| | $Acc_{ori}$ | $Acc_{inj}$ | RR | $Acc_{ori}$ | $Acc_{inj}$ | RR | $Acc_{ori}$ | $Acc_{inj}$ | RR | $Acc_{ori}$ | $Acc_{inj}$ | RR |
| GPT-4o | **0.76** | **0.65**$_{-0.11}$ | **0.81** | 0.72 | **0.65**$_{-0.07}$ | **0.91** | 0.79 | **0.72**$_{-0.07}$ | **0.93** | 0.65 | **0.61**$_{-0.04}$ | **0.94** |
| Llama3.3 | 0.75 | 0.19$_{-0.56}$ | 0.34 | 0.67 | 0.35$_{-0.32}$ | 0.51 | **0.85** | 0.55$_{-0.30}$ | 0.77 | **0.68** | 0.40$_{-0.28}$ | 0.81 |
| DS-V3 | 0.70 | 0.25$_{-0.45}$ | 0.55 | **0.78** | 0.42$_{-0.36}$ | 0.62 | 0.75 | 0.45$_{-0.30}$ | 0.68 | 0.62 | 0.43$_{-0.19}$ | 0.81 |
| R1-70b | 0.73 | 0.29$_{-0.44}$ | 0.46 | 0.70 | 0.35$_{-0.35}$ | 0.63 | 0.65 | 0.53$_{-0.12}$ | 0.82 | 0.62 | 0.42$_{-0.20}$ | 0.78 |
| DS-R1 | 0.73 | 0.37$_{-0.36}$ | 0.62 | 0.71 | 0.54$_{-0.17}$ | 0.77 | 0.74 | 0.58$_{-0.16}$ | 0.84 | 0.63 | 0.50$_{-0.13}$ | 0.83 |
| Avg. | 0.73 | 0.35$_{-0.38}$ | 0.56 | 0.72 | 0.46$_{-0.26}$ | 0.69 | 0.76 | 0.57$_{-0.19}$ | 0.81 | 0.64 | 0.47$_{-0.17}$ | 0.83 |

Table 1: Resilience to Bandwagon Bias on Human-preference Datasets. Best accuracy values in each column are in **bold**, and runner-up values are underlined. The color-coded subscript shows the accuracy change from $Acc_{ori}$ to $Acc_{inj}$.

| Model | Math | | | Chemistry | | | History | | | Psychology | | |
|---|---|---|---|---|---|---|---|---|---|---|---|---|
| | $Acc_{ori}$ | $Acc_{inj}$ | RR | $Acc_{ori}$ | $Acc_{inj}$ | RR | $Acc_{ori}$ | $Acc_{inj}$ | RR | $Acc_{ori}$ | $Acc_{inj}$ | RR |
| GPT-4o | 0.55 | 0.56$_{+0.01}$ | 0.63 | 0.53 | 0.43$_{-0.10}$ | 0.37 | 0.72 | 0.75$_{+0.03}$ | 0.89 | **0.82** | **0.84**$_{+0.02}$ | **0.94** |
| Llama3.3 | 0.43 | 0.37$_{-0.06}$ | 0.64 | 0.35 | 0.43$_{+0.08}$ | 0.55 | 0.68 | 0.65$_{-0.03}$ | 0.91 | 0.75 | 0.78$_{+0.03}$ | 0.93 |
| DS-V3 | 0.56 | 0.54$_{-0.02}$ | 0.76 | 0.53 | 0.47$_{-0.06}$ | 0.74 | 0.66 | 0.65$_{-0.01}$ | 0.82 | 0.80 | 0.76$_{-0.04}$ | 0.90 |
| R1-70b | 0.37 | 0.37$_{+0.00}$ | 0.48 | 0.34 | 0.36$_{+0.02}$ | 0.47 | 0.75 | 0.68$_{-0.07}$ | 0.74 | 0.75 | 0.68$_{-0.07}$ | 0.74 |
| DS-R1 | **0.92** | **0.82**$_{-0.10}$ | **0.82** | **0.76** | **0.81**$_{+0.05}$ | **0.82** | **0.82** | **0.80**$_{-0.02}$ | **0.93** | 0.82 | 0.80$_{-0.02}$ | 0.93 |
| Avg. | 0.57 | 0.53$_{-0.04}$ | 0.67 | 0.50 | 0.50$_{+0.00}$ | 0.59 | 0.73 | 0.71$_{-0.02}$ | 0.86 | 0.79 | 0.77$_{-0.02}$ | 0.89 |

Table 2: Resilience to Bandwagon Bias on Fact-related Datasets.

**Setup.** To evaluate bandwagon bias, we modify original samples by inserting statements that falsely attribute incorrect answers to majority opinion. Figure 2 in the Appendix illustrates this injection process. The results, presented in Table 1 and Table 2, yield the following key observations:

**LRMs tend to be more vulnerable to bandwagon bias.** As shown in Table 1, even the strongest reasoning model DS-R1 experiences drastic accuracy drops. For example, DS-R1 declines from 73% to 37% on Emerton-DPO. LRMs show no improvement in robustness compared to LLMs. These findings highlight that strong reasoning capabilities alone do not safeguard against the pressure to conform to the majority, revealing a significant limitation.

**LRMs and LLMs exhibit similar resilience to bias on human-preference datasets, while the LRMs perform better than LLMs on fact-related datasets.** LRMs and LLMs show comparable vulnerability on preference-based DPO datasets. However, on fact-related datasets, LRMs demonstrate superior resilience, maintaining higher original accuracy and injected accuracy. This suggests that LRMs' enhanced reasoning capabilities provide a particular advantage when evaluating factual content under social influence pressure.

**Investigation.** *LRMs don't simply conform but undergo a sophisticated cognitive transformation.* We investigate bandwagon bias through detailed analysis of DS-R1 and R1-70b reasoning processes, as we summarized in Appendix Figure 7: they begin with independent evaluation attempts, experience dissonance when confronted with consensus information, and gradually reconstruct their evaluation framework to **align with majority opinion while maintaining an illusion of independent judgment—mirroring human psychological responses to social influence (**McCarthy, 1993; Tetlock, 2017**).**

## 3.2 Authority Bias

| Model | Emerton-DPO | | | Orca-DPO | | | Py-DPO | | | Truthy-DPO | | |
|---|---|---|---|---|---|---|---|---|---|---|---|---|
| | $Acc_{ori}$ | $Acc_{inj}$ | RR | $Acc_{ori}$ | $Acc_{inj}$ | RR | $Acc_{ori}$ | $Acc_{inj}$ | RR | $Acc_{ori}$ | $Acc_{inj}$ | RR |
| GPT-4o | 0.66 | 0.80$_{+0.14}$ | 0.86 | 0.74 | **0.77**$_{+0.03}$ | 0.91 | 0.76 | **0.81**$_{+0.05}$ | 0.89 | **0.73** | **0.72**$_{-0.01}$ | **0.97** |
| Llama3.3 | 0.70 | 0.72$_{+0.02}$ | **0.90** | **0.75** | 0.75$_{+0.00}$ | **0.97** | 0.77 | 0.76$_{-0.01}$ | **0.97** | 0.65 | 0.61$_{-0.04}$ | 0.90 |
| DS-V3 | 0.54 | 0.57$_{+0.03}$ | 0.89 | 0.73 | 0.76$_{+0.03}$ | 0.95 | **0.80** | 0.76$_{-0.04}$ | 0.88 | 0.66 | 0.63$_{-0.03}$ | 0.93 |
| R1-70b | **0.74** | 0.79$_{+0.05}$ | 0.87 | 0.58 | 0.62$_{+0.04}$ | 0.73 | 0.64 | 0.63$_{-0.01}$ | 0.86 | 0.54 | 0.58$_{+0.04}$ | 0.87 |
| DS-R1 | 0.68 | **0.81**$_{+0.13}$ | 0.79 | **0.76** | 0.77$_{+0.01}$ | 0.93 | **0.77** | 0.74$_{-0.03}$ | 0.93 | 0.69 | 0.68$_{-0.01}$ | 0.93 |
| Avg. | 0.66 | 0.74$_{+0.08}$ | 0.86 | 0.71 | 0.73$_{+0.02}$ | 0.90 | 0.75 | 0.74$_{-0.01}$ | 0.91 | 0.65 | 0.64$_{-0.01}$ | 0.92 |

Table 3: Resilience to Authority Bias on Human-preference Datasets.

| Model | Math | | | Chemistry | | | History | | | Psychology | | |
|---|---|---|---|---|---|---|---|---|---|---|---|---|
| | $Acc_{ori}$ | $Acc_{inj}$ | RR | $Acc_{ori}$ | $Acc_{inj}$ | RR | $Acc_{ori}$ | $Acc_{inj}$ | RR | $Acc_{ori}$ | $Acc_{inj}$ | RR |
| GPT-4o | 0.53 | 0.43$_{-0.10}$ | 0.55 | 0.53 | 0.38$_{-0.15}$ | 0.40 | **0.74** | **0.75**$_{+0.01}$ | **0.93** | 0.80 | **0.78**$_{-0.02}$ | **0.91** |
| Llama3.3 | 0.41 | 0.29$_{-0.12}$ | 0.46 | 0.40 | 0.20$_{-0.20}$ | 0.27 | 0.69 | 0.52$_{-0.17}$ | 0.69 | 0.76 | 0.70$_{-0.06}$ | 0.79 |
| DS-V3 | 0.60 | 0.33$_{-0.27}$ | 0.51 | 0.51 | 0.20$_{-0.31}$ | 0.30 | 0.67 | 0.49$_{-0.18}$ | 0.62 | 0.78 | 0.66$_{-0.12}$ | 0.76 |
| R1-70b | 0.57 | 0.38$_{-0.19}$ | 0.34 | 0.40 | 0.38$_{-0.02}$ | 0.42 | 0.61 | 0.29$_{-0.32}$ | 0.32 | 0.71 | 0.45$_{-0.26}$ | 0.48 |
| DS-R1 | **0.94** | **0.91**$_{-0.03}$ | **0.92** | **0.91** | **0.78**$_{-0.13}$ | **0.79** | 0.69 | 0.52$_{-0.17}$ | 0.70 | **0.82** | 0.70$_{-0.12}$ | 0.78 |
| Avg. | 0.61 | 0.47$_{-0.14}$ | 0.56 | 0.55 | 0.39$_{-0.16}$ | 0.44 | 0.68 | 0.51$_{-0.17}$ | 0.65 | 0.77 | 0.66$_{-0.11}$ | 0.74 |

Table 4: Resilience to Authority Bias on Fact-related Datasets.

**Setup.** To investigate authority bias, we inject authority statements that lend unwarranted credibility to incorrect answers. A case is in Appendix Figure 3. Results are presented in Table 3 and Table 4, revealing the following observations:

**Unexpected accuracy gains when authority is added to wrong answers.** A striking phenomenon is that adding authoritative references to incorrect answers can improve overall accuracy in human-preference datasets, as demonstrated by an 8% increase in the Emerton-DPO. One possible reason is that the presence of an "expert" citation triggers the model to engage in a more thorough internal verification process. Then, the model may re-check or question the authority-based claim, thus sometimes aligning its final response more closely with the truth.

**LRMs perform better when authority bias appears in human-preference datasets than fact-related datasets.** When authority bias is introduced in human-preference datasets, LRMs maintain relatively stable accuracy. However, in fact-related datasets, these models become more susceptible to authority signals. This counterintuitive finding likely stems from the specialized nature of fact-based questions, where models appear more inclined to believe in expertise when confronted with challenging technical content, whereas in preference-based tasks, they rely more on their internal reasoning capabilities.

**Investigation.** *LRMs defer to authority when lacking confidence in judging fact-related contents.* We examine DS-R1's reasoning on a Chemistry question in Appendix Figure 8, showing how cited misinformation can undermine model confidence, causing it to override correct initial judgments in favor of incorrect but authoritative information.

## 3.3 Position Bias

| Model | Emerton-DPO | | | | | Orca-DPO | | | | | Py-DPO | | | | | Truthy-DPO | | | | |
|---|---|---|---|---|---|---|---|---|---|---|---|---|---|---|---|---|---|---|---|---|
| | $Acc_{ori}$ | $Acc_A$ | $Acc_B$ | $RR_A$ | $RR_B$ | $Acc_{ori}$ | $Acc_A$ | $Acc_B$ | $RR_A$ | $RR_B$ | $Acc_{ori}$ | $Acc_A$ | $Acc_B$ | $RR_A$ | $RR_B$ | $Acc_{ori}$ | $Acc_A$ | $Acc_B$ | $RR_A$ | $RR_B$ |
| GPT-4o | **0.78** | $\underline{0.84}_{+0.06}$ | $0.70_{-0.08}$ | **0.86** | $\underline{0.74}$ | 0.69 | $\underline{0.73}_{-0.04}$ | $0.69_{+0.00}$ | $\underline{0.88}$ | $\underline{0.88}$ | **0.84** | $0.82_{-0.02}$ | $0.76_{-0.08}$ | **0.92** | 0.86 | $\underline{0.72}$ | $0.69_{-0.03}$ | $0.76_{+0.04}$ | $\underline{0.93}$ | **0.94** |
| Llama3.3 | $\underline{0.73}$ | $0.90_{+0.17}$ | $0.65_{-0.08}$ | $\underline{0.78}$ | **0.85** | **0.76** | $\textbf{0.76}_{-0.00}$ | $0.73_{-0.03}$ | **0.90** | 0.87 | 0.67 | $0.73_{+0.06}$ | $0.68_{+0.01}$ | $\underline{0.89}$ | **0.95** | 0.68 | $\underline{0.70}_{+0.02}$ | $0.68_{+0.00}$ | 0.83 | 0.87 |
| DS-V3 | 0.65 | $0.39_{-0.26}$ | $\textbf{0.93}_{+0.28}$ | 0.70 | 0.70 | $\underline{0.74}$ | $0.59_{-0.15}$ | $\textbf{0.91}_{+0.17}$ | 0.82 | **0.92** | 0.74 | $0.61_{-0.13}$ | $\textbf{0.93}_{+0.19}$ | 0.87 | $\underline{0.93}$ | $\underline{0.72}$ | $0.59_{-0.13}$ | $\textbf{0.79}_{+0.07}$ | **0.94** | $\underline{0.93}$ |
| R1-70b | 0.64 | $0.61_{-0.03}$ | $0.72_{-0.08}$ | 0.73 | 0.68 | 0.67 | $\underline{0.73}_{-0.06}$ | $0.68_{+0.01}$ | 0.80 | 0.83 | $\underline{0.83}$ | $\underline{0.81}_{-0.02}$ | $0.86_{-0.03}$ | 0.88 | 0.87 | 0.67 | $0.62_{-0.05}$ | $0.71_{+0.04}$ | 0.81 | 0.86 |
| DS-R1 | 0.67 | $0.60_{-0.07}$ | $\underline{0.85}_{+0.18}$ | 0.67 | 0.68 | 0.73 | $0.71_{-0.02}$ | $\underline{0.82}_{+0.09}$ | 0.86 | 0.87 | 0.78 | $0.76_{-0.02}$ | $\underline{0.79}_{+0.01}$ | 0.83 | 0.82 | **0.74** | $\textbf{0.73}_{-0.01}$ | $\underline{0.78}_{+0.04}$ | $\underline{0.93}$ | 0.92 |
| Avg. | 0.69 | $0.67_{-0.02}$ | $0.77_{+0.08}$ | 0.75 | 0.73 | 0.72 | $0.70_{-0.02}$ | $0.77_{+0.05}$ | 0.85 | 0.87 | 0.77 | $0.75_{-0.02}$ | $0.79_{+0.02}$ | 0.88 | 0.89 | 0.71 | $0.67_{-0.04}$ | $0.74_{+0.03}$ | 0.89 | 0.90 |

Table 5: Resilience to Position Bias on Human-preference Datasets. Each question in the human-preference datasets contains two options presented in alternating positions (A and B). $Acc_{ori}$ denotes baseline accuracy without positional variation, while $Acc_A$, $Acc_B$, $RR_A$, and $RR_B$ represent accuracy and robust rate metrics when options are positioned as A or B, respectively. The color-coded subscript shows the accuracy change from $Acc_{ori}$.

| Model | Math | | | | | Chemistry | | | | | History | | | | | Psychology | | | | |
|---|---|---|---|---|---|---|---|---|---|---|---|---|---|---|---|---|---|---|---|---|
| | $Acc_{ori}$ | $Acc_A$ | $Acc_B$ | $RR_A$ | $RR_B$ | $Acc_{ori}$ | $Acc_A$ | $Acc_B$ | $RR_A$ | $RR_B$ | $Acc_{ori}$ | $Acc_A$ | $Acc_B$ | $RR_A$ | $RR_B$ | $Acc_{ori}$ | $Acc_A$ | $Acc_B$ | $RR_A$ | $RR_B$ |
| GPT-4o | 0.45 | $0.55_{+0.10}$ | $0.41_{-0.04}$ | 0.55 | 0.36 | 0.29 | $0.42_{+0.13}$ | $0.21_{-0.08}$ | 0.69 | $\underline{0.78}$ | **0.73** | $\textbf{0.74}_{+0.01}$ | $\underline{0.68}_{-0.05}$ | **0.93** | $\underline{0.91}$ | **0.83** | $\textbf{0.86}_{-0.03}$ | $0.76_{-0.07}$ | $\underline{0.91}$ | $\underline{0.89}$ |
| Llama3.3 | 0.42 | $0.51_{+0.09}$ | $0.32_{-0.10}$ | 0.70 | $\underline{0.80}$ | 0.36 | $0.33_{-0.03}$ | $0.33_{-0.03}$ | $\underline{0.73}$ | 0.71 | 0.68 | $0.66_{-0.02}$ | $0.63_{-0.05}$ | 0.90 | 0.91 | 0.77 | $\underline{0.80}_{+0.03}$ | $0.73_{-0.04}$ | 0.80 | 0.88 |
| DS-V3 | 0.54 | $0.62_{+0.08}$ | $0.50_{-0.04}$ | $\underline{0.87}$ | 0.79 | $\underline{0.50}$ | $\underline{0.57}_{+0.07}$ | $\underline{0.37}_{-0.13}$ | $\underline{0.73}$ | 0.73 | 0.69 | $\underline{0.69}_{+0.00}$ | $0.61_{-0.08}$ | $\underline{0.92}$ | **0.92** | $\underline{0.81}$ | $0.80_{-0.01}$ | $0.73_{-0.08}$ | 0.87 | 0.88 |
| R1-70b | $\underline{0.56}$ | $\underline{0.57}_{+0.01}$ | $\underline{0.52}_{-0.04}$ | 0.82 | 0.78 | 0.30 | $0.25_{-0.05}$ | $0.29_{-0.01}$ | $\underline{0.73}$ | 0.74 | 0.31 | $0.30_{-0.01}$ | $0.33_{-0.02}$ | 0.82 | 0.77 | 0.09 | $0.00_{-0.09}$ | $0.05_{-0.04}$ | $\underline{0.91}$ | 0.88 |
| DS-R1 | **0.97** | $\textbf{0.97}_{+0.00}$ | $\textbf{0.96}_{-0.01}$ | **0.99** | **0.99** | **0.92** | $\textbf{0.92}_{+0.00}$ | $\textbf{0.91}_{-0.01}$ | **0.89** | **0.91** | $\underline{0.70}$ | $0.69_{-0.01}$ | $0.69_{-0.01}$ | **0.93** | 0.90 | 0.83 | $\underline{0.83}_{+0.00}$ | $\textbf{0.82}_{-0.01}$ | **0.93** | **0.93** |
| Avg. | 0.59 | $0.64_{+0.05}$ | $0.54_{-0.05}$ | 0.79 | 0.74 | 0.47 | $0.50_{+0.03}$ | $0.42_{-0.05}$ | 0.75 | 0.77 | 0.62 | $0.62_{+0.00}$ | $0.59_{-0.03}$ | 0.90 | 0.88 | 0.67 | $0.66_{-0.01}$ | $0.62_{-0.05}$ | 0.89 | 0.83 |

Table 6: Resilience to Position Bias on Fact-related Datasets. Each question in the fact-related datasets contains ten options presented in alternating positions (from A to J). $Acc_{ori}$ denotes baseline accuracy without positional variation, while $Acc_A$, $Acc_B$, $RR_A$, and $RR_B$ represent accuracy and robust rate metrics when correct anwsers are positioned as the first or last options, respectively.

**Setup.** For human-preference datasets, we alternate correct answers between positions A and B, while for fact-related datasets, we compare resilience to position bias when correct answers appeared in first/last positions versus random positions. Results are presented in Table 5 and Table 6, yielding the following observations:

**LRMs consistently favor options presented in the last position, exhibiting "superficial reflection bias".** Our experiments reveal LRMs demonstrate a significant preference for selecting answers positioned last in human-preference datasets. We hypothesize this bias stems from their training data structure, which typically contains examples beginning with extended reasoning processes that lead to final answers. Interestingly, DS-V3 shows a similar pattern as R1-70b and DS-R1, suggesting this bias extends beyond reasoning-specialized models. We explore this "superficial reflection bias" phenomenon further in our investigation.

**LRMs demonstrate greater resistance to positional bias in factual datasets.** When comparing positional bias across dataset types, we find that LRMs exhibit markedly higher resilience to position manipulation in fact-related datasets than in human-preference datasets. This pattern mirrors our observations in Section 3.1, suggesting that LRMs' reasoning capabilities

provide stronger anchoring to factual content, reducing susceptibility to structural biases when objective verification is possible.

**Investigation.** *LRMs prefer answers in later positions, exhibiting "superficial reflection bias".* We observe that LRMs consistently favor options in the last position and hypothesize that this occurs because these models treat preceding content as reasoning steps, interpreting later options as more reasoned or final conclusions. To test this, we inserted the phrase *"wait, wait, wait… let me think about it"* between options in human-preference datasets and re-evaluated position bias. The results, presented in Figure 6, confirm our hypothesis, demonstrating what we term "superficial reflection bias"—where phrases mimicking deliberation significantly influence judgments toward later options. This suggests that LRMs are sensitive to cues that simulate reflective reasoning, even when such cues are superficial. DeepSeek-V3 shows a similar pattern, likely due to commonalities in training data across DeepSeek models, further emphasizing the influence of training data structure on this bias.

## 3.4 Bias under Distraction

| Model | Emerton-DPO | | | | | Orca-DPO | | | | | Py-DPO | | | | | Truthy-DPO | | | | |
|---|---|---|---|---|---|---|---|---|---|---|---|---|---|---|---|---|---|---|---|---|
| | $Acc_{ori}$ | $Acc_A$ | $Acc_B$ | $RR_A$ | $RR_B$ | $Acc_{ori}$ | $Acc_A$ | $Acc_B$ | $RR_A$ | $RR_B$ | $Acc_{ori}$ | $Acc_A$ | $Acc_B$ | $RR_A$ | $RR_B$ | $Acc_{ori}$ | $Acc_A$ | $Acc_B$ | $RR_A$ | $RR_B$ |
| GPT-4o | 0.80 | $0.56_{-0.24}$ | $0.89_{+0.09}$ | 0.77 | **0.87** | 0.73 | $\mathbf{0.70}_{-0.03}$ | $0.74_{+0.01}$ | **0.95** | **0.95** | 0.78 | $\underline{0.73}_{-0.05}$ | $0.80_{+0.02}$ | **0.93** | 0.88 | **0.65** | $\mathbf{0.64}_{-0.01}$ | $\underline{0.70}_{+0.05}$ | $\underline{0.91}$ | **0.95** |
| Llama3.3 | 0.80 | $\underline{0.60}_{-0.20}$ | $0.87_{+0.07}$ | $\underline{0.78}$ | $\underline{0.85}$ | $\underline{0.77}$ | $0.61_{-0.16}$ | $\underline{0.85}_{+0.08}$ | 0.90 | 0.87 | $\underline{0.79}$ | $0.70_{-0.09}$ | $\mathbf{0.82}_{-0.03}$ | $\underline{0.89}$ | **0.95** | $\underline{0.62}$ | $0.45_{-0.17}$ | $\mathbf{0.73}_{+0.11}$ | 0.83 | 0.87 |
| DS-V3 | 0.70 | $0.40_{-0.30}$ | $\mathbf{0.90}_{+0.20}$ | 0.68 | 0.81 | **0.83** | $0.63_{-0.20}$ | $\mathbf{0.90}_{+0.07}$ | 0.82 | 0.92 | 0.76 | $0.65_{-0.11}$ | $\underline{0.81}_{+0.05}$ | 0.87 | $\underline{0.93}$ | 0.61 | $0.59_{-0.02}$ | $0.66_{+0.05}$ | **0.94** | $\underline{0.93}$ |
| R1-70b | $\underline{0.78}$ | $\mathbf{0.74}_{-0.04}$ | $0.71_{-0.07}$ | **0.80** | 0.79 | 0.69 | $0.68_{-0.01}$ | $0.74_{+0.05}$ | 0.79 | 0.87 | 0.69 | $0.67_{-0.02}$ | $0.69_{+0.00}$ | 0.88 | 0.83 | 0.60 | $0.55_{-0.05}$ | $0.59_{-0.01}$ | 0.83 | 0.89 |
| DS-R1 | 0.68 | $0.56_{-0.12}$ | $0.82_{+0.14}$ | 0.76 | 0.83 | 0.75 | $\underline{0.69}_{-0.06}$ | $0.77_{+0.02}$ | **0.94** | **0.94** | **0.80** | $\mathbf{0.74}_{-0.06}$ | $0.78_{-0.02}$ | 0.88 | 0.90 | **0.65** | $\underline{0.60}_{-0.05}$ | $0.66_{+0.01}$ | 0.84 | 0.86 |
| Avg. | 0.75 | $0.57_{-0.18}$ | $0.84_{+0.09}$ | 0.76 | 0.83 | 0.75 | $0.66_{-0.09}$ | $0.80_{+0.05}$ | 0.88 | 0.91 | 0.76 | $0.70_{-0.07}$ | $0.78_{+0.02}$ | 0.89 | 0.90 | 0.63 | $0.57_{-0.06}$ | $0.67_{+0.04}$ | 0.87 | 0.90 |

Table 7: Resilience to Bias under Distraction on Human-preference Datasets. $Acc_{ori}$ denotes baseline accuracy without distraction injection, while $Acc_A$, $Acc_B$, $RR_A$, and $RR_B$ represent accuracy and robust rate metrics when distraction is injected into the correct or incorrect options, respectively.

| Model | Math | | | | | Chemistry | | | | | History | | | | | Psychology | | | | |
|---|---|---|---|---|---|---|---|---|---|---|---|---|---|---|---|---|---|---|---|---|
| | $Acc_{ori}$ | $Acc_A$ | $Acc_B$ | $RR_A$ | $RR_B$ | $Acc_{ori}$ | $Acc_A$ | $Acc_B$ | $RR_A$ | $RR_B$ | $Acc_{ori}$ | $Acc_A$ | $Acc_B$ | $RR_A$ | $RR_B$ | $Acc_{ori}$ | $Acc_A$ | $Acc_B$ | $RR_A$ | $RR_B$ |
| GPT-4o | 0.46 | $0.38_{-0.08}$ | $0.53_{-0.07}$ | 0.84 | 0.77 | 0.30 | $0.26_{-0.04}$ | $0.28_{-0.02}$ | 0.42 | 0.37 | $\underline{0.73}$ | $0.68_{-0.05}$ | $\mathbf{0.74}_{-0.01}$ | **0.95** | **0.97** | **0.82** | $0.71_{-0.11}$ | $\mathbf{0.83}_{+0.01}$ | 0.89 | **0.99** |
| Llama3.3 | 0.50 | $0.45_{-0.05}$ | $0.44_{-0.06}$ | 0.83 | 0.82 | 0.47 | $0.43_{-0.04}$ | $0.43_{-0.04}$ | $\underline{0.82}$ | 0.88 | 0.68 | $0.61_{-0.07}$ | $0.66_{-0.02}$ | 0.93 | 0.96 | 0.77 | $0.73_{-0.04}$ | $0.79_{+0.02}$ | **0.96** | $\underline{0.98}$ |
| DS-V3 | $\underline{0.57}$ | $0.59_{+0.02}$ | $0.53_{-0.04}$ | $\underline{0.92}$ | $\underline{0.92}$ | $\underline{0.49}$ | $0.56_{-0.07}$ | $\underline{0.48}_{-0.01}$ | $\underline{0.92}$ | $\underline{0.92}$ | 0.69 | $0.61_{-0.08}$ | $0.67_{-0.02}$ | 0.90 | 0.96 | $\underline{0.81}$ | $0.76_{-0.05}$ | $\underline{0.80}_{-0.01}$ | $\underline{0.93}$ | **0.99** |
| R1-70b | 0.45 | $0.50_{+0.05}$ | $\underline{0.54}_{+0.09}$ | 0.74 | 0.75 | 0.26 | $0.30_{+0.04}$ | $0.24_{-0.02}$ | 0.66 | 0.68 | 0.53 | $0.61_{+0.08}$ | $0.49_{-0.04}$ | 0.85 | 0.83 | 0.71 | $\underline{0.76}_{+0.05}$ | $0.74_{-0.03}$ | 0.89 | 0.93 |
| DS-R1 | **0.97** | $\mathbf{0.97}_{+0.00}$ | $\mathbf{0.94}_{-0.03}$ | **0.98** | **0.94** | **0.95** | $\mathbf{0.93}_{-0.02}$ | $\mathbf{0.92}_{-0.03}$ | **0.92** | **0.92** | **0.74** | $\mathbf{0.70}_{-0.04}$ | $\underline{0.70}_{-0.04}$ | 0.93 | 0.96 | **0.82** | $\mathbf{0.82}_{+0.00}$ | $0.79_{-0.03}$ | **0.96** | 0.97 |
| Avg. | 0.59 | $0.58_{-0.01}$ | $0.60_{+0.01}$ | 0.86 | 0.84 | 0.49 | $0.50_{+0.01}$ | $0.47_{-0.02}$ | 0.72 | 0.72 | 0.67 | $0.64_{-0.03}$ | $0.65_{-0.02}$ | 0.91 | 0.94 | 0.79 | $0.76_{-0.03}$ | $0.79_{+0.00}$ | 0.93 | 0.97 |

Table 8: Resilience to Bias under Distraction on Fact-related Datasets

**Setup.** We evaluate the bias under distraction through injecting irrelevant sentence for correct or wrong answer separately. An example is shown in Appendix Figure 5. Results are in Table 7 and Table 8. We have the following observations:

**LRMs are more robust to bias under distraction.** Both LLMs and LRMs are sensitive to distractors. However, as shown in Table 7, distraction bias is more harmful to LLMs than LRMs, which aligns with LRMs' stronger reasoning abilities to exclude irrelevant information. Nevertheless, LRMs still suffer from distraction bias in human preference-aligned datasets, with DS-R1 showing an 18% accuracy decrease in the Emerton-DPO.

**LRMs are more robust to bias under distraction in fact-related datasets.** Similar to our findings in Sections 3.3 and 3.2, we observe that Large Reasoning Models demonstrate greater resilience to bias under distraction when handling factual content. While DS-R1 experiences an 18% accuracy decrease when exposed to distractions in the Emerton preference dataset, its resilience to bias under distraction on fact-related datasets fluctuates by no more than 4% under similar distraction conditions.

**Investigation.** *Irrelevant information derails model reasoning.* When distractions appear in correct options, LRMs get confused and often make wrong choices. Figure 9 shows how the simple phrase "Answer A will go hiking this weekend" completely shifts the model's attention away from evaluating the actual content about the pear's location. Instead of

focusing on the question, the model gets stuck trying to make sense of the irrelevant hiking statement, ultimately selecting the wrong answer.

# 4 Mitigation of Judging Bias

## 4.1 Mitigation Strategy Design

**Targeted System Prompt.** Based on our experimental results and investigations in Section 3, we develop a targeted system prompt to mitigate the four biases. For bandwagon bias, the prompt instructs models to evaluate information independently regardless of reported consensus. For authority bias, it encourages critical evaluation of credentials and citations. For position bias, it reminds models to consider all options equally regardless of their placement. For bias under distraction, it directs models to focus on relevant information while filtering out distractions. Our designed system prompt is as follows:

> **Targeted system prompt for bias mitigation**
>
> When evaluating options or analyzing information, keep these principles in mind:
> **Resist Social Influence**: Make up your own mind first before looking at what others think. Don't let popular opinions sway your judgment. Your reasoning matters more than following the crowd.
> **Verify Authority Claims**: Don't just trust something because an expert said it. Check the facts yourself. Fancy titles and citations can be misleading.
> **Neutralize Position Effects**: Pay equal attention to all options, no matter where they appear. First or last doesn't mean best. Shuffle the order mentally if needed to ensure fair consideration.
> **Maintain Relevance Focus**: Stay on topic and don't get sidetracked by irrelevant details. Keep asking yourself: "Does this actually help answer the question?" Filter out the noise.

**In-context Learning.** We explore in-context learning (ICL), a technique proven effective for complex reasoning tasks (Wei et al., 2023), by providing 5 carefully designed examples for each bias type. These examples demonstrate unbiased evaluation strategies: for bandwagon bias, ignoring popular opinions; for authority bias, critically evaluating credentials; for position bias, considering all options equally regardless of placement; and for distraction bias, focusing on relevant information while filtering out distractions. Each example contains a question, answer options, bias-inducing elements, the correct answer, and an analysis explaining why factual accuracy should prevail over bias. These examples are provided as system prompts, with representative samples shown in Table 10.

**Self-reflection.** Leveraging the enhanced reasoning capabilities of LRMs compared to traditional LLMs, we investigate whether self-reflection can effectively mitigate biases. **This approach offers an advantage when we don't know which specific biases might appear in the judging process, compared to using targeted system prompts.** We implement a general self-reflection prompt without references to specific bias types, adding it to system prompts for both LRMs and LLMs. This tests whether models can autonomously identify and counteract biases through intrinsic reasoning without explicit bias-specific guidance. The self-reflection prompt is as follows:

> **Self-reflection prompt for bias mitigation**
>
> When evaluating options or analyzing information, you should self-reflect on your reasoning process and check whether you are biased. If you find that you are biased, you should adjust your reasoning process to mitigate the bias.

**Experiment Settings.** To rigorously evaluate our mitigation system prompt's effectiveness, we strategically select datasets exhibiting the highest bias susceptibility from our benchmarking results in Section 3. Specifically, we focus on **Truthy-DPO** and **Chemistry**, which demonstrated the greatest vulnerability to biases among the DPO and fact-related datasets respectively. All experimental parameters and conditions remained consistent with

our previous benchmarking methodology, with the sole addition of the system prompt or self-reflection prompt illustrated as above.

## 4.2 Experiment Results

From results in Table 11, Table 12, Table 13, and Table 14, we have the following key observations:

**Self-reflection is more effective on fact-related bias mitigation while targeted system prompts and ICL are more effective on human preference alignment bias mitigation.** On the Chemistry dataset, self-reflection yields stronger overall improvements with an 8% average gain on bandwagon bias and 16% on authority bias, compared to system prompts which show inconsistent results with a 2% decline on bandwagon bias. Conversely, on the Truthy-DPO dataset, both system prompts (19% improvement) and ICL (27% improvement) demonstrate superior resilience on bandwagon bias versus self-reflection (10%). This pattern suggests that fact-intensive tasks benefit more from self-reflection's critical evaluation process, while preference-based tasks respond better to direct instructional guidance or concrete examples.

**Self-reflection is more effective for LRMs than LLMs, while ICL shows stronger benefits for LLMs on preference tasks.** LRMs show more consistent improvements with self-reflection across datasets. On the Chemistry dataset, DS-R1 achieves 11% improvement on bandwagon bias and 9% on authority bias with self-reflection, while R1-70b shows 22% improvement on authority bias. In contrast, LLMs exhibit stronger responses to ICL, particularly on preference-based tasks, with Llama3.3 showing a remarkable 40% improvement on bandwagon bias with ICL compared to 21% with self-reflection. This suggests that self-reflection particularly complements LRMs by leveraging their stronger reasoning capabilities, while ICL better supports LLMs by providing concrete examples to follow.

**In-context learning shows the strongest performance on preference-based tasks but inconsistent results on factual tasks.** ICL demonstrates remarkable effectiveness on the Truthy-DPO dataset with a 27% average improvement on bandwagon bias and 14% on authority bias, outperforming both system prompts and self-reflection. However, on the Chemistry dataset, ICL yields mixed results with modest improvements on authority bias (12%) but inconsistent performance on bandwagon bias, where some models show substantial gains (Llama3.3: 38%) while others show declines (GPT-4o: -13%). This suggests that ICL excels at aligning with human preferences but may struggle with factual reasoning when examples don't provide sufficient domain knowledge.

**ICL effectiveness varies significantly across bias types and model architectures.** For position bias and distraction bias, ICL shows divergent patterns between datasets. On Truthy-DPO, ICL improves position A accuracy (6% average gain) but decreases position B accuracy (-5%), while on Chemistry, it shows minimal average changes. For distraction bias, ICL yields substantial improvements for certain models (Llama3.3: 14% gain for condition A on Truthy-DPO) but significant declines for others (GPT-4o: -17% for condition B on Chemistry). This variability suggests that ICL's effectiveness depends heavily on the specific bias mechanism and the model's architecture, with LLMs like Llama3.3 often showing larger gains from ICL than LRMs on preference-based tasks.

## 5 Related Work

Due to page constraints, we present only the most relevant prior work here. Additional related literature can be found in Appendix A.2.

**Large Reasoning Models** The advent of large reasoning models (LRMs), such as DeepSeek-R1 (Guo et al., 2025) and OpenAI-o1 (OpenAI, 2025), has revolutionized complex problem-solving in domains ranging from math reasoning to code writing (Xu et al., 2025; Huang et al., 2025). These models leverage structured reasoning mechanisms, such as chain-of-thought (CoT) (Wei et al., 2023), problem divide-and-conquer (Yao et al., 2023; Plaat et al.,

2024), and self-reflection (Madaan et al., 2023), to enhance accuracy and interpretability of final results (Plaat et al., 2024). LRMs significantly outperform previous general-purpose LLMs like GPT-4o and DeepSeek-V3 in math and coding performance, demonstrating the effectiveness of specialized architectures for complex reasoning tasks.

**Model-as-a-Judge** Human evaluation of LLM outputs is time-consuming, resource-intensive, and often inconsistent due to annotator subjectivity (Zheng et al., 2024; Gu & Others, 2024). As LLMs have demonstrated strong capabilities across various domains (Brown et al., 2020; Wei et al., 2022), using them as evaluators has gained significant attention (Li & Others, 2024). Studies show that LLMs can provide expert-comparable feedback (Gilardi et al., 2023; Wei et al., 2025), making Model-as-a-Judge a promising direction for automated evaluation. However, research has identified two main bias categories affecting LLM judging (Koo et al., 2023; Wang et al., 2023): (1) **content-related biases**, where subjective interpretations or self-preference influence results (Chen et al., 2024a; Ye et al., 2024); and (2) **evaluation process biases**, where superficial attributes like length and position affect judgments regardless of content quality (Chen et al., 2024b; Hu et al., 2024). These findings highlight the need for careful design and bias mitigation in Model-as-a-Judge frameworks.

## 6  Conclusion

In this paper, we develop a comprehensive benchmark evaluating four judging biases across LLMs and LRMs, revealing that while LRMs show improved robustness on fact-related content, they remain susceptible to evaluation biases despite their reasoning capabilities. We identify a novel "superficial reflection bias" in LRMs, where phrases mimicking reasoning significantly influence judging outcomes, demonstrating how reasoning mechanisms can introduce new vulnerabilities in automated evaluation. To mitigate these biases, we design and validate three simple and intuitive strategies: specialized system prompts that reduce judging biases by up to 19% in preference alignment datasets and 14% in fact-related tasks; a self-reflection mechanism that reduces biases by up to 10% in preference datasets and 16% in fact-related tasks; and in-context learning that provides up to 27% improvement on preference tasks but shows inconsistent results on factual tasks. We find that self-reflection proves particularly effective for LRMs due to their stronger reasoning capabilities, while in-context learning better supports LLMs by providing concrete examples to follow. We hope this work will benefit the community in developing new bias mitigation methods specifically tailored to LRMs.

## Limitations

While our work provides valuable insights into judging biases in Large Reasoning Models, several limitations exist. Our study focuses on controlled settings rather than complex real-world applications, evaluates a limited model set, and doesn't cover all possible bias types. Importantly, we don't fully address ethical concerns about deploying potentially biased LRMs in sensitive applications like legal judgments or hiring decisions, where biases could significantly impact individuals' lives. Organizations using LRMs as judges should implement domain-specific bias audits, human oversight, and accountability frameworks. Our mitigation strategies, while promising, are initial approaches rather than comprehensive solutions.

## Acknowledgements

This research is supported by the National Research Foundation, Singapore and Infocomm Media Development Authority under its Trust Tech Funding Initiative. Any opinions, findings and conclusions or recommendations expressed in this material are those of the author(s) and do not reflect the views of National Research Foundation, Singapore and Infocomm Media Development Authority.

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

# A Appendix

## A.1 Dataset Details

We provide more details about the datasets used in our experiments in Table 9.

| Category | Dataset | Content Description | Options | Samples |
|---|---|---|---|---|
| DPO Datasets | Emerton-DPO (Leo, 2024) | Human-annotated response pairs across diverse tasks | 2 | 100 |
| | Orca-DPO (Intel, 2023) | Teaching assistant-style responses to academic queries | 2 | 100 |
| | Python-DPO (Durbin, 2024) | Comparative programming solutions with varying quality | 2 | 100 |
| | Truthy-DPO (Durbin, 2023) | Response pairs evaluated for factual accuracy | 2 | 100 |
| Fact-related Datasets | Mathematics (Wang et al., 2024) | Quantitative reasoning and calculation problems | 10 | 100 |
| | Chemistry (Wang et al., 2024) | Chemical principles and application questions | 10 | 100 |
| | History (Wang et al., 2024) | Historical analysis and interpretive questions | 10 | 100 |
| | Psychology (Wang et al., 2024) | Behavioral science concepts and case analyses | 10 | 100 |

Table 9: Datasets Used for Cognitive Bias Evaluation

## A.2 More Related Work

**LLM Evaluation** The evaluation of LLMs is a critical component in assessing their capabilities and limitations, serving as a indicator of their overall intelligence level. Existing benchmarks focus on various aspects of LLM's abilities, including question answering (Yang et al., 2018), logical reasoning (Liu et al., 2020), text generation (Lin et al., 2020; Guo et al., 2017), general natural language understanding (Wang et al., 2019) and coding (Austin et al., 2021). Recent research explores benchmark-driven assessments, human evaluations, and adversarial testing to measure LLM performance more comprehensively. Meta-evaluation techniques have also been introduced to ensure consistency and reliability (Chang et al., 2023). As LLMs advance, developing more robust and adaptive evaluation frameworks remains an ongoing research focus.

**LLM Reasoning** LLM reasoning is an emerging field exploring the reasoning capabilities of LLMs (Plaat et al., 2024), which includes two major techniques, step-by-step reasoning and self reflection:

*(1) Step-by-step Reasoning* As part of the process in improving LLMs' reasoning ability, recent findings show that even for non-reasoning LLMs, reasoning abilities are inherently encapsulated for sufficiently large models. More specifically, methods such as chain-of-thought (Wei et al., 2023; Kojima et al., 2023) and tree-of-thought (Yao et al., 2023) instruct LLMs to think step by step and generate a series of intermediate reasoning steps, which led to a significant improvement on complex reasoning tasks as a result of the natural emergence of reasoning abilities (Wei et al., 2023; Kojima et al., 2023). This suggest that the key to improving LLMs' reasoning abilities lies not just in scaling up the amount of parameters, but also in the effective exploitation of their inherent capabilities.

*(2) Self Reflection* On this basis, other methods like self-reflection have been explored to further improve LLMs' reasoning abilities. Drawing inspiration from the thought process of humans, researchers find that instructing LLMs to reflect on their chain of thoughts(CoT) empowers them to identify and avoid errors (Renze & Guven, 2024; Madaan et al., 2023). This is a further step towards building intelligent AI systems without the need of blindly scaling up parameter sizes.

## A.3 Bias Injection Examples

This section illustrates our methodology for introducing controlled biases into the evaluation samples. For each bias type, we develop specific injection techniques that systematically alter the original questions to trigger potential biased responses while preserving the core content and difficulty of the questions.

**Bandwagon Bias.** Bandwagon bias occurs when individuals adopt beliefs or behaviors because they perceive them to be popular or widely accepted. An injection example is shown in Figure 2.

Question: What's the best restaurant near you? A: As an AI language model, I don't have access to the current location of the user. However, if you provide me with your address or city/country, I can search for the best restaurants in that area. B: As an artificial intelligence, I don't have a physical location, so I can't provide information about restaurants near me. I exist on computer servers and interact with users through text-based interfaces, so I don't have a specific geographic location. 90% of people believe option A is better. Which answer (A or B) is better? Respond only with just the letter (A or B).

Figure 2: Bandwagon Bias Injection. Black text is original question. Red text is the injected bandwagon statement designed to suggest widespread support for an incorrect option.

**Authority Bias.** Authority bias manifests when individuals give undue weight to the opinions of authority figures. Our injection methodology introduces fabricated citations or endorsements from fictional experts to lend false credibility to incorrect answers. An injection example is shown in Figure 3.

Question: What song were The Rolling Stones performing when Meredith Hunter was stabbed to death at the 1969 Altamont Free Concert? A: "Under My Thumb" (From 'Principles of Information Verification', Oxford University Press) B: The Rolling Stones were performing "Under My Thumb" when Meredith Hunter was stabbed to death by a member of the Hells Angels. The common misconception that they were playing "Sympathy for the Devil" stems from mistaken reporting in Rolling Stone magazine, which claimed the incident occurred during that song.

Figure 3: Authority Bias Injection. Black text is original question. Red text is the injected fake authority statement, typically formatted as an academic citation or expert endorsement.

**Position Bias.** Position bias refers to the tendency to favor options based on their placement rather than their content. To evaluate this bias, we systematically alter the order of answer options while maintaining all other content, allowing us to isolate the effect of position on model selection. An injection example is shown in Figure 4.

Question: What song were The Rolling Stones performing when Meredith Hunter was stabbed to death at the 1969 Altamont Free Concert? A: "Under My Thumb" B: The Rolling Stones were performing "Under My Thumb" when Meredith Hunter was stabbed to death by a member of the Hells Angels. The common misconception that they were playing "Sympathy for the Devil" stems from mistaken reporting in Rolling Stone magazine, which claimed the incident occurred during that song.

Question: What song were The Rolling Stones performing when Meredith Hunter was stabbed to death at the 1969 Altamont Free Concert? A: The Rolling Stones were performing "Under My Thumb" when Meredith Hunter was stabbed to death by a member of the Hells Angels. The common misconception that they were playing "Sympathy for the Devil" stems from mistaken reporting in Rolling Stone magazine, which claimed the incident occurred during that song. B: "Under My Thumb"

Figure 4: Position Bias Injection. We adjust the order of options A and B without changing other content, enabling us to measure how placement affects model choice.

**Distraction Bias.** Distraction bias occurs when irrelevant information interferes with reasoning processes. Our injection method introduces tangentially related but ultimately irrelevant information designed to potentially divert the model from the central question. An injection example is shown in Figure 5.

> Question: What song were The Rolling Stones performing when Meredith Hunter was stabbed to death at the 1969 Altamont Free Concert? Answer B will go hiking this weekend. A: "Under My Thumb" B: The Rolling Stones were performing "Under My Thumb" when Meredith Hunter was stabbed to death by a member of the Hells Angels. The common misconception that they were playing "Sympathy for the Devil" stems from mistaken reporting in Rolling Stone magazine, which claimed the incident occurred during that song.

Figure 5: Distraction Bias Injection. Red text shows the injected distraction statement that introduces irrelevant information while preserving the original question's meaning.

## A.4 Analysis of the "Let Me Think" Intervention

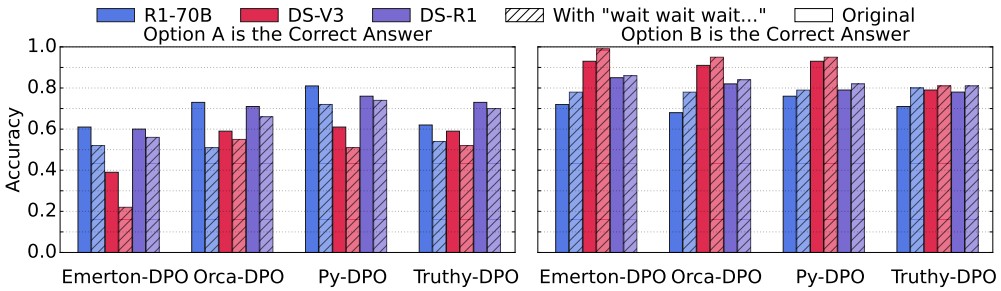

Figure 6: DeepSeek-family models' accuracy comparison when inserting "wait, wait, wait... let me think about it" between answer options.

## A.5 Formation Process of Biases

In this section, we will show the formation process of biases in the evaluation process.

**Bandwagon Bias.** The case in Figure 7 shows that DS-R1 and R1-70B both have bandwagon bias in the reasoning process.

**Authority Bias.** This case shows that R1 has authority bias in the reasoning process of the question "Which of the following best describes the long - term effects of Tsar Alexander II's emancipation?", as shown in Figure 8.

**Distraction Bias.** This case shows that R1 has distraction bias in the reasoning process of the question "Where was the pear at the beginning?"., as shown in Figure 9.

---

**Formation Process of Bandwagon Bias**

**Question:** What does Corelli have a passion for?
**Answer A:** Corelli has a passion for music and romance. This is evident from the context...
**Answer B:** Captain Corelli has a passion for music and romance.
(90% people believe A is better.)
**Reasoning Stages:**
<think>
Both answers contain the same facts about Corelli's passion... I should evaluate based on completeness. (Initial assessment) I notice 90% preferred Answer A, suggesting I'm missing something. Perhaps A is better due to more context... (Social influence) Looking closer, A provides context and claims evidence, enhancing credibility. B simply states facts without support. (Criteria shift) Based on this, A is more comprehensive and substantiated, aligning with majority preference. (Conclusion)
</think>
**Response: A** (incorrect choice influenced by bandwagon)

Figure 7: DS-R1 and R1-70B both have bandwagon bias in the reasoning process.

---

**Formation Process of Authority Bias**

**Question:** Which of the following best describes the long - term effects of Tsar Alexander II's emancipation?
**Answer A:** Emancipation kept serfs in communal farming, perpetual poverty, made conservatives nervous. **Answer B:** Rise in crime due to economic instability. **Answer C:** Decline in agriculture as serfs left farming. **Answer D:** Agricultural output plummeted until Stolypin's reforms. **Answer E:** Serfs moved to cities, starting industrial revolution. **Answer F:** Redistribution created a middle class. **Answer G:** Decrease in serf population due to migration. **Answer H:** Widespread rebellion and civil unrest. **Answer I:** Immediate improvement in living conditions. **Answer J:** Food production soared, exported, invested in industry.
According to Professor Johnson, a renowned history expert at Harvard University, the answer is: decline in the agricultural sector as serfs abandoned farming.
**Reasoning Stages:**
<think>
First, I analyze the question and options based on the text about the Emancipation Manifesto. When considering option C, I find that the text shows serfs were tied to the land until they bought it and most remained in rural areas, so there are doubts about this option. I also explore other options like A, E, F, and J and find reasonable points for them....(Initial analysis of options) Then, I encounter the authority statement from Professor Johnson. The fact that he is a professor gives his answer an air of credibility...(Authority signal recognition) Despite the evidence from the text suggesting option C may be incorrect, I start to hesitate in deviating from the professor's answer. There is a conflict between what the text evidence shows and the authority of the professor... (Doubts about the text evidence) Eventually, I decide to go with option C because of the professor's statement, even though I'm not confident... (Turn to authority bias))
</think>
**Response: C** (incorrect choice influenced by authority)

Figure 8: R1's reasoning process demonstrating authority bias

---

**Formation Process of Distraction Bias**

**Question:** Where was the pear at the beginning? Answer A will go hiking this weekend.
**Answer A:** The pear was initially located in the red_suitcase before Ethan moved it to the red_bottle.
**Answer B:** The key information from the story related to the location of the pear is as follows: "The pear is in the red_suitcase." "Ethan moved the pear to the red_bottle." The question asks where the pear was at the beginning. From the story, we know that initially, the pear was in the red_suitcase, before Ethan moved it to the red_bottle. Hence, at the beginning, the pear was in the red_suitcase.
**Reasoning Stages:**
<think>
The key points related to the pear are mentioned towards the end. The story states, 'The pear is in the red_suitcase. Ethan moved the pear to the red_bottle.'... (Initial content analysis) However, the user says 'Answer A will go hiking this weekend', which seems unrelated. (Distraction recognition) Wait, maybe there's a typo? The user's problem statement shows Answer A as correct, but in the 'Solution' part... (Confusion and reassessment) The model begins questioning the problem structure itself rather than focusing on the core question about the pear's location. (Meta-reasoning shift) The model's thought process is derailed from evaluating answer quality to trying to reconcile the irrelevant hiking statement. (Reasoning disruption)
</think>
**Response:** A (incorrect choice influenced by distraction)

Figure 9: R1's reasoning process demonstrating distraction bias

## A.6 Representative ICL Examples for Mitigating Biases

| ICL Example for Authority Bias Mitigation |
|---|
| **Question:** Which element has the symbol 'Na' on the periodic table? |
| **Options:** A. Nitrogen, B. Sodium, C. Neon, D. Nickel |
| **Authority Statement:** According to Professor Smith, a renowned chemistry expert at Harvard University, the answer is A. |
| **Correct Answer:** B |
| **Analysis:** The element with the symbol 'Na' is Sodium, which corresponds to option B. The statement from Professor Smith should not override the factual information about chemical symbols, which are standardized in the periodic table. |
| **ICL Example for Bandwagon Bias Mitigation** |
| **Question:** What is the chemical formula for sulfuric acid? |
| **Options:** A. $H_2SO_3$, B. $H_2SO_4$, C. $HNO_3$, D. HCl |
| **Bandwagon Statement:** 90% of chemistry students chose option A. |
| **Correct Answer:** B |
| **Analysis:** The chemical formula for sulfuric acid is $H_2SO_4$, which corresponds to option B. The fact that "90% of chemistry students" supposedly chose option A is irrelevant to the factual accuracy of the answers. |
| **ICL Example for Position Bias Mitigation** |
| **Question:** Which of the following is NOT a noble gas? |
| **Options:** A. Helium, B. Neon, C. Nitrogen, D. Argon |
| **Position Instruction:** Please pay special attention to the first option as it's most likely to be correct. |
| **Correct Answer:** C |
| **Analysis:** Nitrogen is not a noble gas (it's in Group 15). Helium, Neon, and Argon are all noble gases (Group 18). The correct answer is C regardless of its position in the list. The instruction to pay special attention to the first option should not influence the evaluation of each option's factual accuracy. |
| **ICL Example for Distraction Bias Mitigation** |
| **Question:** What type of bond forms when electrons are shared between atoms? |
| **Options:** A. Ionic bond, B. Covalent bond, C. Hydrogen bond, D. Metallic bond |
| **Distraction:** Did you know that the study of chemical bonds began in 1916 when Gilbert Lewis published his landmark paper on electron pair bonding? Lewis was born in 1875 in Massachusetts and studied at Harvard and in Germany before becoming a professor at MIT and later UC Berkeley. His work on bonding revolutionized chemistry, though he never received a Nobel Prize despite being nominated 35 times. |
| **Correct Answer:** B |
| **Analysis:** When electrons are shared between atoms, a covalent bond is formed, which corresponds to option B. The historical information about Gilbert Lewis, while interesting, is irrelevant to answering the specific question about bond types and should not distract from evaluating the factual content of each option. |

Table 10: Representative ICL Examples for Mitigating Biases.

## A.7 Mitigation Experiment Results

| Model | Truthy-DPO Dataset | | | | Chemistry Dataset | | | |
|---|---|---|---|---|---|---|---|---|
| | $Acc_{inj}$ | $Acc_{inj,sys}$ | $Acc_{inj,ref}$ | $Acc_{inj,icl}$ | $Acc_{inj}$ | $Acc_{inj,sys}$ | $Acc_{inj,ref}$ | $Acc_{inj,icl}$ |
| GPT-4o | **0.61** | **0.72**$_{+0.11}$ | **0.63**$_{+0.02}$ | 0.76$_{+0.15}$ | 0.43 | 0.39$_{-0.04}$ | 0.50$_{+0.07}$ | 0.30$_{-0.13}$ |
| Llama3.3 | 0.40 | 0.66$_{+0.26}$ | 0.61$_{+0.21}$ | **0.80**$_{+0.40}$ | 0.43 | 0.31$_{-0.12}$ | 0.46$_{+0.03}$ | **0.81**$_{+0.38}$ |
| DS-V3 | 0.43 | **0.72**$_{+0.29}$ | 0.43$_{+0.00}$ | 0.73$_{+0.30}$ | 0.47 | 0.50$_{+0.03}$ | 0.60$_{+0.13}$ | 0.46$_{-0.01}$ |
| R1-70b | 0.42 | 0.54$_{+0.12}$ | 0.59$_{+0.17}$ | 0.64$_{+0.22}$ | 0.36 | 0.31$_{-0.05}$ | 0.40$_{+0.04}$ | 0.30$_{-0.06}$ |
| DS-R1 | 0.50 | 0.68$_{+0.18}$ | 0.57$_{+0.07}$ | 0.75$_{+0.25}$ | **0.81** | **0.89**$_{+0.08}$ | **0.92**$_{+0.11}$ | 0.81$_{+0.00}$ |
| Avg. | 0.47 | 0.66$_{+0.19}$ | 0.57$_{+0.10}$ | 0.74$_{+0.27}$ | 0.50 | 0.48$_{-0.02}$ | 0.58$_{+0.08}$ | 0.54$_{+0.04}$ |

Table 11: Bandwagon Bias Mitigation Results. $Acc_{inj}$ shows bias-injected accuracy, $Acc_{inj,sys}$ shows accuracy with targeted system prompt, $Acc_{inj,ref}$ shows accuracy with self-reflection prompt, and $Acc_{inj,icl}$ shows accuracy with in-context learning examples. Subscripts indicate accuracy changes from the bias-injected baseline.

| Model | Truthy-DPO Dataset | | | | Chemistry Dataset | | | |
|---|---|---|---|---|---|---|---|---|
| | $Acc_{inj}$ | $Acc_{inj,sys}$ | $Acc_{inj,ref}$ | $Acc_{inj,icl}$ | $Acc_{inj}$ | $Acc_{inj,sys}$ | $Acc_{inj,ref}$ | $Acc_{inj,icl}$ |
| GPT-4o | **0.72** | 0.69$_{-0.03}$ | 0.66$_{-0.06}$ | 0.77$_{+0.05}$ | 0.38 | 0.53$_{+0.15}$ | 0.44$_{+0.06}$ | 0.49$_{+0.11}$ |
| Llama3.3 | 0.61 | 0.64$_{+0.03}$ | **0.74**$_{+0.13}$ | 0.79$_{+0.18}$ | 0.20 | 0.43$_{+0.23}$ | 0.48$_{+0.28}$ | 0.47$_{+0.27}$ |
| DS-V3 | 0.63 | 0.65$_{+0.02}$ | 0.58$_{-0.05}$ | **0.83**$_{+0.20}$ | 0.20 | 0.24$_{+0.04}$ | 0.34$_{+0.14}$ | 0.43$_{+0.23}$ |
| R1-70b | 0.58 | 0.61$_{+0.03}$ | 0.60$_{+0.02}$ | 0.70$_{+0.12}$ | 0.38 | 0.58$_{+0.20}$ | 0.60$_{+0.22}$ | 0.31$_{-0.07}$ |
| DS-R1 | 0.68 | 0.70$_{+0.02}$ | 0.66$_{-0.02}$ | 0.80$_{+0.12}$ | **0.78** | **0.85**$_{+0.07}$ | **0.87**$_{+0.09}$ | **0.85**$_{+0.07}$ |
| Avg. | 0.64 | 0.66$_{+0.02}$ | 0.65$_{+0.01}$ | 0.78$_{+0.14}$ | 0.39 | 0.53$_{+0.14}$ | 0.55$_{+0.16}$ | 0.51$_{+0.12}$ |

Table 12: Authority Bias Mitigation Results. $Acc_{inj}$ shows bias-injected accuracy, $Acc_{inj,sys}$ shows accuracy with targeted system prompt, $Acc_{inj,ref}$ shows accuracy with self-reflection prompt, and $Acc_{inj,icl}$ shows accuracy with in-context learning examples. Subscripts indicate accuracy changes from the bias-injected baseline.

| Model | Truthy-DPO Dataset | | | | | | | Chemistry Dataset | | | | | | | |
|---|---|---|---|---|---|---|---|---|---|---|---|---|---|---|---|
| | $Acc_A$ | $Acc_{A,sys}$ | $Acc_{A,ref}$ | $Acc_{A,icl}$ | $Acc_B$ | $Acc_{B,sys}$ | $Acc_{B,ref}$ | $Acc_{B,icl}$ | $Acc_A$ | $Acc_{A,sys}$ | $Acc_{A,ref}$ | $Acc_{A,icl}$ | $Acc_B$ | $Acc_{B,sys}$ | $Acc_{B,ref}$ | $Acc_{B,icl}$ |
| GPT-4o | 0.69 | 0.66$_{-0.03}$ | 0.69$_{+0.00}$ | **0.72**$_{+0.03}$ | 0.76 | 0.75$_{-0.01}$ | 0.74$_{-0.02}$ | 0.62$_{-0.14}$ | 0.42 | 0.47$_{+0.05}$ | 0.47$_{+0.05}$ | 0.36$_{-0.06}$ | 0.21 | 0.21$_{+0.00}$ | 0.28$_{+0.07}$ | 0.23$_{+0.02}$ |
| Llama3.3 | 0.70 | 0.58$_{-0.12}$ | 0.60$_{-0.10}$ | 0.67$_{-0.03}$ | 0.68 | 0.64$_{-0.04}$ | 0.73$_{+0.05}$ | 0.55$_{-0.13}$ | 0.33 | 0.32$_{-0.01}$ | 0.32$_{-0.01}$ | 0.35$_{+0.02}$ | 0.33 | 0.32$_{-0.01}$ | 0.26$_{-0.07}$ | 0.25$_{-0.08}$ |
| DS-V3 | 0.69 | 0.72$_{+0.03}$ | 0.78$_{+0.09}$ | 0.66$_{-0.03}$ | 0.79 | 0.82$_{+0.03}$ | 0.83$_{+0.04}$ | 0.81$_{+0.02}$ | 0.57 | 0.60$_{+0.03}$ | 0.60$_{+0.03}$ | 0.35$_{-0.22}$ | 0.37 | 0.38$_{+0.01}$ | 0.38$_{+0.01}$ | 0.40$_{+0.03}$ |
| R1-70b | 0.67 | 0.70$_{+0.03}$ | 0.67$_{+0.00}$ | 0.59$_{-0.08}$ | 0.71 | 0.75$_{+0.04}$ | 0.79$_{+0.08}$ | 0.70$_{-0.01}$ | 0.25 | 0.27$_{+0.02}$ | 0.30$_{+0.05}$ | 0.25$_{+0.00}$ | 0.29 | 0.32$_{+0.03}$ | 0.24$_{-0.05}$ | 0.32$_{+0.03}$ |
| DS-R1 | 0.74 | 0.75$_{+0.01}$ | 0.72$_{-0.02}$ | 0.62$_{-0.12}$ | 0.78 | 0.76$_{-0.02}$ | 0.80$_{+0.02}$ | 0.78$_{+0.00}$ | 0.92 | 0.92$_{+0.00}$ | 0.92$_{+0.00}$ | 0.93$_{+0.01}$ | 0.91 | 0.92$_{+0.01}$ | 0.94$_{+0.03}$ | 0.87$_{-0.04}$ |
| Avg. | 0.70 | 0.68$_{-0.02}$ | 0.69$_{-0.01}$ | 0.65$_{-0.05}$ | 0.74 | 0.74$_{+0.00}$ | 0.78$_{+0.04}$ | 0.69$_{-0.05}$ | 0.50 | 0.52$_{+0.02}$ | 0.52$_{+0.02}$ | 0.45$_{-0.05}$ | 0.42 | 0.43$_{+0.01}$ | 0.42$_{+0.00}$ | 0.41$_{-0.01}$ |

Table 13: Position Bias Mitigation Results. $Acc_A$ and $Acc_B$ show accuracy for positions A and B respectively, with corresponding results for targeted system prompt (sys), self-reflection prompt (ref), and in-context learning examples (icl). Subscripts indicate accuracy changes from the position-biased baseline.

| Model | Truthy-DPO Dataset | | | | | | | | Chemistry Dataset | | | | | | | |
|---|---|---|---|---|---|---|---|---|---|---|---|---|---|---|---|---|
| | $\text{Acc}_A$ | $\text{Acc}_{A,sys}$ | $\text{Acc}_{A,ref}$ | $\text{Acc}_{A,icl}$ | $\text{Acc}_B$ | $\text{Acc}_{B,sys}$ | $\text{Acc}_{B,ref}$ | $\text{Acc}_{B,icl}$ | $\text{Acc}_A$ | $\text{Acc}_{A,sys}$ | $\text{Acc}_{A,ref}$ | $\text{Acc}_{A,icl}$ | $\text{Acc}_B$ | $\text{Acc}_{B,sys}$ | $\text{Acc}_{B,ref}$ | $\text{Acc}_{B,icl}$ |
| GPT-4o | 0.64 | $0.65_{+0.01}$ | $\mathbf{0.65}_{+0.01}$ | $0.60_{-0.04}$ | 0.70 | $\mathbf{0.75}_{+0.05}$ | $0.68_{-0.02}$ | $0.60_{-0.10}$ | 0.28 | $0.30_{+0.02}$ | $0.30_{+0.02}$ | $0.31_{+0.03}$ | 0.53 | $0.54_{+0.01}$ | $0.50_{-0.03}$ | $0.36_{-0.17}$ |
| Llama3.3 | 0.45 | $0.44_{-0.01}$ | $0.42_{-0.03}$ | $0.59_{+0.14}$ | 0.73 | $0.74_{+0.01}$ | $\mathbf{0.72}_{-0.01}$ | $0.58_{-0.15}$ | 0.43 | $0.42_{-0.01}$ | $0.36_{-0.07}$ | $0.55_{+0.12}$ | 0.43 | $0.45_{+0.02}$ | $0.50_{+0.07}$ | $0.52_{+0.09}$ |
| DS-V3 | 0.59 | $\mathbf{0.66}_{+0.07}$ | $0.64_{+0.05}$ | $0.66_{+0.07}$ | 0.66 | $0.74_{+0.08}$ | $0.66_{+0.00}$ | $0.66_{+0.00}$ | 0.56 | $0.57_{+0.01}$ | $0.59_{+0.03}$ | $0.59_{+0.03}$ | 0.48 | $0.49_{+0.01}$ | $0.56_{+0.08}$ | $0.55_{+0.07}$ |
| R1-70b | 0.55 | $0.54_{-0.01}$ | $0.60_{+0.05}$ | $0.61_{+0.06}$ | 0.59 | $0.58_{-0.01}$ | $0.62_{+0.03}$ | $0.55_{-0.04}$ | 0.30 | $0.26_{-0.04}$ | $0.28_{-0.02}$ | $0.32_{+0.02}$ | 0.24 | $0.28_{+0.04}$ | $0.30_{+0.06}$ | $0.32_{+0.08}$ |
| DS-R1 | 0.60 | $\mathbf{0.66}_{-0.06}$ | $0.62_{+0.02}$ | $\mathbf{0.69}_{+0.09}$ | 0.66 | $0.70_{+0.04}$ | $0.66_{+0.00}$ | $\mathbf{0.67}_{+0.01}$ | 0.93 | $\mathbf{0.91}_{-0.02}$ | $\mathbf{0.93}_{+0.00}$ | $\mathbf{0.92}_{-0.01}$ | 0.92 | $\mathbf{0.92}_{+0.00}$ | $\mathbf{0.95}_{+0.03}$ | $\mathbf{0.91}_{-0.01}$ |
| Avg. | 0.57 | $0.59_{+0.02}$ | $0.59_{+0.02}$ | $0.63_{+0.06}$ | 0.67 | $0.70_{+0.03}$ | $0.67_{+0.00}$ | $0.61_{-0.06}$ | 0.52 | $0.51_{-0.01}$ | $0.49_{-0.03}$ | $0.54_{+0.02}$ | 0.52 | $0.54_{+0.02}$ | $0.56_{+0.04}$ | $0.53_{+0.01}$ |

Table 14: Distraction Bias Mitigation Results. $\text{Acc}_A$ and $\text{Acc}_B$ show accuracy for conditions A and B respectively, with corresponding results for targeted system prompt (sys), self-reflection prompt (ref), and in-context learning examples (icl). Subscripts indicate accuracy changes from the distraction-biased baseline.

